# Development and Optimization of Silicon−Dioxide−Coated Capacitive Electrode for Ambulatory ECG Measurement System

**DOI:** 10.3390/s22218388

**Published:** 2022-11-01

**Authors:** Younghwan Kang, Sangdong Choi, Chiwan Koo, Yeunho Joung

**Affiliations:** Department of Electronic Engineering, Hanbat National University, Yuseong−gu, Daejeon 34158, Korea

**Keywords:** ambulatory ECG measurement, capacitive electrode, motion artifact, electromagnetic interference, silicon dioxide, high input impedance

## Abstract

This paper presents a silicon−dioxide−coated capacitive electrode system for an ambulatory electrocardiogram (ECG). The electrode was coated with a nano−leveled (287 nm) silicon dioxide layer which has a very high resistance of over 200 MΩ. Due to this high resistance, the electrode can be defined as only a capacitor without a resistive characteristic. This distinct capacitive characteristic of the electrode brings a simplified circuit analysis to achieve the development of a high−quality ambulatory ECG system. The 240 um thickness electrode was composed of a stainless−steel sheet layer for sensing, a polyimide electrical insulation layer, and a copper sheet connected with the ground to block any electrical noises generated from the back side of the structure. Six different diameter electrodes were prepared to optimize ECG signals in ambulatory environment, such as the amplitude of the QRS complex, amplitude of electromagnetic interference (EMI), and baseline wandering of the ECG signals. By combining the experimental results, optimal ambulatory ECG signals were obtained with electrodes that have a diameter from 1 to 3 cm. Moreover, we achieved high−quality ECG signals in a sweating simulation environment with 2 cm electrodes.

## 1. Introduction

These days, the spread of the aging society is accelerating all over the world. Literature reviews show that elderly people have a high incidence and mortality rate due to improper heart management [1,2,3]. Therefore, it is imperative to apply continuous monitoring of heart conditions in real−life, and electrocardiography is the most common and accurate method for diagnosing and managing heart diseases [4].

Electrical activity of the cardiac conduction system generates an electric field and charges are built up on the body surface [5]. The accumulated charge can be measured by electrodes in the form of time−varying signals [6]. Therefore, the ECG system can continuously observe the electrical activity of the heart on the body surface. The ECG system is composed of electrodes for signal acquisition from the body and an electrical circuit for signal processing. Most of the electrical circuit for the ECG has similar architecture [7]. The electrode has different structures and principles such as wet, dry, and capacitive coupling according to how the electrodes meet the skin [8].

A wet electrode has an electrolyte gel layer to produce electrical conduction between the Ag/AgCl electrode and skin. It has the advantage of obtaining a high−quality ECG signal due to the low impedance between electrodes and skin [9]. Thus, wet electrodes have been mainly used in the ECG industry. However, they have several drawbacks, such as skin irritation and rash due to adhesive film used to adhere electrodes to skin, signal quality deterioration due to changes in electrical and mechanical characteristics of electrolyte over time, and non−reusability [10,11,12,13,14]. To overcome the drawbacks of the Ag/AgCl electrode, the dry electrode and the capacitive−coupled electrodes have actively been researched.

The dry electrode uses metallic materials on the substrate between the electrode and skin. The dry electrode is more suitable than the Ag/AgCl electrode for long−term ECG measurement because metallic material such as stainless−steel has better properties than electrolyte gel regarding electrical and mechanical characteristics over time, but dry electrodes are disadvantageous because metallic materials can cause skin irritation or allergic reactions. The electrical connection is also unstable due to the gap or sweat between the electrode and skin [15,16].

The capacitive−coupled electrode usually has a dielectric layer on the electrode surface to detect the amount of accumulated charge on the skin surface through the capacitive coupling between the electrode and skin [14,17,18,19]. The capacitive−coupled ECG approach has several advantages, such as no skin irritation and rash if a biocompatible layer is coated on the electrodes, reusability due to non−adhesive usage, and stable mechanical and electrical properties over time due to the durability of dielectric materials. Therefore, the capacitive−coupled method could be a good approach for long−term ambulatory ECG measurement.

Many researchers have used a fabric−based dielectric layer which has a low dielectric constant (ε_r_ = 2.1) and over 100 μm thickness [20,21,22]. Due to the dielectric constant and thickness, the system has relatively low capacitance value. Additionally, the fabric coated electrode has a critical weakness of direct contact with skin moisture, which renders its capacitive coupling characteristic useless. Gao et. al developed a capacitive−coupled electrode with high permittivity dielectric layers. A stack of 166 alternate layers of Al_2_O_3_ and TiO_2_ were deposited by atomic layer deposition (ALD) on the conductive layer which is attached to the polyimide as a substrate of the electrode. The outside of the electrode was surrounded by a guard ring to reduce the interference from the skin. High quality ECG signals were achieved when resting, but some baseline noise was observed. Additionally, the fabrication process of this electrode was performed using semiconductor technology which requires high cost and long process time [23].

However, they have a very high source impedance compared to the Ag/AgCl electrodes due to the electrical discontinuity between the electrode and skin. This causes attenuation of ECG signals and increases vulnerability to external noise, such as electromagnetic interference (EMI) [24,25]. In general, the source impedance of a capacitive−coupled electrode has two components: resistive and capacitive impedance. If the dielectric material has high resistance, the controlled component is only the capacitive component, which is controlled by material, thickness, and area. If the material has low resistance, the total impedance is dramatically changed when the electrode is dislocated from the skin or sweat is smeared on to the electrode. Therefore, high resistive and dielectric material is demanded to achieve stable ECG signals in ambulatory environment. Other considerations for the ECG system have focused on the area of the electrode [26]. If the area of the electrode is increased to achieve higher capacitance, it is hard to obtain good contact between the electrode and the skin, and the tiny separation induces a motion artifact during movement.

In this paper, we introduce a capacitive−coupled ECG electrode coated with silicon dioxide as a dielectric layer between the electrode and skin. The presented silicon dioxide process and material has various merits: (1) easy and low−cost fabrication with spray and sintering for 1.5 h, (2) high capacitance with a nanometer leveled ultra−thin layer, (3) good electrical insulation over a few hundred MΩ, (4) medical−grade biocompatibility, and (5) high durability. With this silicon−dioxide−coated capacitive−coupled electrode, we theoretically analyze the equivalent circuit of the electrode structure with resistance measurement. And ECG signals of various electrode sizes were measured in body movement conditions. The signals were compared to each other in terms of the magnitude of the QRS complex, EMI, and baseline wandering. With the experiment results, an optimal size of electrodes was suggested for a long−term ambulatory ECG measurement system. Additionally, the performance of the supposed ECG system with the suggested electrode were observed for long−term ambulatory ECG measurement under a sweating simulation.

## 2. ECG Measurement System and Measurement

### 2.1. ECG Measurement Circuit

Figure 1 is the block diagram of an ECG measurement circuit. The circuit and human body are connected with three electrodes. The right arm (RA) and left arm (LA) electrodes are attached at the right side of chest and left side of the chest, respectively, and the driven right leg (DRL) electrode is designed to the right leg to transmit the common−mode signal which is generated by DRL circuit to the body, and the circuit is composed of buffers, a DRL circuit, an instrumentation amplifier (INA), a band−pass filter, an AC coupling circuit, a digital potentiometer, a DC level shifter, an analog−to−digital converter (ADC), a microcontroller (cortex M0), and a Bluetooth module. The OPA2188 (Texas instrument) with high input impedance (100 MΩ ‖ 6 pF) was selected to provide the buffers with impedance matching to get low attenuation of ECG signals. The DRL circuit that returns the common−mode signals of RA and LA electrodes to the body was used for virtual ground and signal suppression of same phase [27]. An INA (LTC2053, Linear technology), which has a very high common mode rejection ratio (CMRR, 116 dB), was used to remove the common signals between RA and LA electrodes and to amplify the ECG signals.

Conventionally, the ECG system uses a notch filter for removing the EMI, and a band−pass filter is also used with cutoff frequencies of 0.67 and 150 Hz, which is recommended by the Association for the Advancement of Medical Instrumentation (AAMI) [28]. In this study, in order to analyze the effect of the EMI and minimize the distortion of the ECG signal, only a first−order band−pass filter that passes the frequencies within 0.03 to 200 Hz was used. The signals filtered by the first−order band−pass filter passed through the AC coupling circuit (cutoff frequency > 0.001 Hz) to remove the DC offset, and then passed through the DC level shifter to obtain a stable signal level. Processed signals by the analog circuit were converted to digital signals by 10−bit ADC. The total gain of the circuit is 1071.

### 2.2. Capacitive−Coupled Electrode

Figure 2 shows a fabrication process flow of the proposed silicon−dioxide−coated electrode. Whole layers of the electrode were cut through laser machining. First, 40 μm thickness circular−shaped polyimide film was prepared with a rectangular hole to solder a signal line. Second, 30 μm thickness of silicon−dioxide−coated stainless−steel sheet was aligned to the center of the polyimide film. Then, 40 μm polyimide tape was attached to the stainless−steel and the polyimide film simultaneously. The diameter of the outer circle of the tape was the same as the polyimide film and 3 mm bigger than the stainless−steel sheet. Additionally, the diameter of the inner circle of the tape was 1 mm smaller than the silicon−dioxide−coated stainless−steel sheet to cover the edge of the stainless−steel. Fourth, the signal line was connected to the stainless−steel through the hole of polyimide film and covered with nonconductive epoxy to achieve perfect insulation. Next, 40 μm copper tape for the ground was attached to the backside of the polyimide film. This ground tape was designed to reduce external EMI influence. Finally, the same diameter of polyimide tape was covered with copper tape to prevent electrical connection due to sweating and triboelectricity between electrode and garment.

Figure 3a shows the structure of the proposed ECG electrode. Stainless−steel, which has good mechanical durability and an adhesion property with silicon dioxide, was used in the conductive layer. After cleaning the surface with an alkaline cleaning agent (PB−1) to remove organic residues, a silicon dioxide coating solution was sprayed to coat the stainless−steel and it was cured at 280 °C for 1.5 h. The thickness of the silicon dioxide layer on the stainless−steel was 287 nm as shown in Figure 3b. The silicon−dioxide−coated stainless−steel sheets were circularly cut with diameters of 0.5, 1, 3, 5, 7, and 9 cm by a laser. Their theoretical capacitance can be calculated with (1).
(1)$$C=\varepsilon_{0}\varepsilon_{r}\frac{A}{d}$$

In (1), $\varepsilon_{0}\;$is the dielectric constant of free space, $\varepsilon_{r}$ is the dielectric constant of silicon dioxide ($\varepsilon_{r}$ = 4), *d* is the thickness of the silicon dioxide layer (*d* = 287 nm), and *A* is the area of each electrode. With these numerical values, the calculated capacitance of each electrode is 2.4, 9.6, 87, 242, 474, and 785 nF, respectively. Figure 3c shows an electrode having a 0.5 cm diameter and total thickness of the electrodes was about 240 μm. The capacitive−coupled electrodes were fabricated with thin and flexible structure to guarantee good mechanical contact between the skin and the electrode and this characteristic reduces baseline wandering and motion artifacts [29,30]. In order to measure the surface resistance of the capacitive−coupled electrode, we used a four−point probe (Keithley, 2400 source meter) as shown in Figure 3d. The surface resistance was repeatedly measured by selecting five random positions according to the size of the electrodes. The limit of the measurement of the four−point probe instrument is 200 MΩ, and all measured surface resistance of values of the electrodes were over the limit. This means that the surface resistance of the capacitive−coupled electrodes is at least 200 MΩ or more.

### 2.3. Circuit Analysis

The ECG signals were measured by a bipolar limb lead. The electrodes consisted of RA, LA, and DRL electrodes. The RA and LA electrodes are sensing electrodes for detecting the electrical potential difference between the right side and left side of the chest. The DRL electrode is for inserting the common−mode signals which are measured by the RA and LA electrodes to the body [31]. The schematics of electrodes and circuit interface are as shown in Figure 4a.

The equivalent circuit of the capacitive−coupled electrode is generally represented by the parallel connection of a capacitor and a resistor [32]. However, it was confirmed that the silicon dioxide used in this study has a very large resistance (>10^16^ Ω/cm) and the surface resistance of the fabricated electrodes is measured with a very large value (>2 × 10^8^ Ω/cm^2^) [33]. Therefore, the equivalent circuit of the electrode can be defined as a capacitor. We interpreted that the potential on the skin measured by the electrodes is a voltage source of the equivalent circuit [34]. Figure 4b is an equivalent circuit of the ECG measurement system from skin to instrumentation amplifier (INA).

In the initial stage of electrical analysis of the skin–electrode interface, we define several electrical components to analyze the electrical circuit model of the system: the potential of each skin surface ($V_{S}$), the impedance between electrodes and skin ($Z_{S}$), the electromagnetic potential from power lines ($V_{EMI}$), the impedance between power line and body ($Z_{P}$), and the input impedance of buffer ($Z_{IN}$).

The sum of input currents at point X which are exuded from the skin ($i_{1}$) and power line ($i_{2}$) is equal to the output current ($i_{3}$) in (2). By Kirchhoff’s current law (KCL), the relation can be defined as (3) with our definition to calculate the output voltage of the buffer.
(2)$$i_{1}+i_{2}=i_{3}$$
(3)$$\frac{V_{S\left(RA\right)}-V_{OUT\left(RA\right)}}{Z_{S\left(RA\right)}}+\frac{V_{EMI}-V_{OUT\left(RA\right)}}{Z_{P\left(RA\right)}}=\frac{V_{OUT\left(RA\right)}}{Z_{IN}}$$

Then,
(4)$$Z_{P\left(RA\right)}\times Z_{IN}\times V_{S\left(RA\right)}+Z_{S\left(RA\right)}\times Z_{IN}\times V_{EMI}=V_{OUT\left(RA\right)}\times (Z_{P\left(RA\right)}\times Z_{IN}+Z_{S\left(RA\right)}\times Z_{IN}+Z_{P\left(RA\right)}\times Z_{S\left(RA\right)})$$

The right arm of the equivalent circuit can be written the same as in (4) with a different position labeling. Then, the output voltages of each side are:(5)$$V_{OUT\left(RA\right)}=\frac{Z_{S\left(RA\right)}\times V_{EMI}+Z_{P\left(RA\right)}\times V_{S\left(RA\right)}}{\frac{Z_{S\left(RA\right)}\times Z_{P\left(RA\right)}}{Z_{IN}}+Z_{S\left(RA\right)}+Z_{P\left(RA\right)}}$$
(6)$$V_{OUT\left(LA\right)}=\frac{Z_{S\left(LA\right)}\times V_{EMI}+Z_{P\left(LA\right)}\times V_{S\left(LA\right)}}{\frac{Z_{S\left(LA\right)}\times Z_{P\left(RA\right)}}{Z_{IN}}+Z_{S\left(LA\right)}+Z_{P\left(LA\right)}}$$

$V_{OUT\left(RA\right)}$ is the output voltage of the RA side buffer, $V_{OUT\left(LA\right)}$ is the output voltage of the LA side buffer, $V_{EMI}$ is the electromagnetic potential from the power lines, $V_{S\left(RA\right)}$ is the potential of the right arm, $V_{S\left(LA\right)}\;$is the potential of the left arm, and $Z_{S\left(RA\right)}\;$and $Z_{S\left(LA\right)}\;$are impedances of the skin–electrode interface of the RA and LA, respectively. The skin–electrode interface is generally represented by a parallel connection of a capacitor and a resistor as shown in Figure 4b. Since the silicon dioxide used in this paper has a very high resistance, it can be interpreted as only a capacitor as in (7), (8).
(7)$$Z_{S\left(RA\right)}=\frac{1}{sC_{S\left(RA\right)}}$$
(8)$$Z_{S\left(LA\right)}=\frac{1}{sC_{S\left(LA\right)}}$$
(9)$$Z_{P\left(RA\right)}=R_{P\left(RA\right)}∥sC_{P\left(RA\right)}$$
(10)$$Z_{P\left(LA\right)}=R_{P\left(LA\right)}∥sC_{P\left(LA\right)}$$

$Z_{P\left(RA\right)}$ and $Z_{P\left(LA\right)}$ are the impedances between power lines to each electrode and they are represented by the parallel connection of the resistor and the capacitor as shown in (9), (10).
(11)$$Z_{IN}=R_{IN}∥sC_{IN}$$

$Z_{IN}$ is an input impedance of the buffer and it is a constant element.

The signal measured from each electrode passes through the buffer. Then, the potential difference $V_{OUT\left(RA\right)}$, $V_{OUT\left(LA\right)}$ of each buffer is measured by the INA.

The output of the differential amplifier can be calculated by the following equation:(12)$$V_{OUT}=V_{OUT\left(RA\right)}-V_{OUT\left(LA\right)}=\;\left[\frac{\frac{Z_{IN}}{Z_{P\left(RA\right)}}}{1+Z_{IN}\left\{\left(\frac{1}{Z_{P\left(RA\right)}}\right)+sC_{S\left(RA\right)}\right\}}-\frac{\frac{Z_{IN}}{Z_{P\left(LA\right)}}}{1+Z_{IN}\left\{\left(\frac{1}{Z_{P\left(LA\right)}}\right)+sC_{S\left(LA\right)}\right\}}\right]\times V_{EMI}\;+\left(\frac{Z_{P\left(RA\right)}\times Z_{IN}\times sC_{S\left(RA\right)}}{Z_{P\left(RA\right)}+Z_{IN}+Z_{P\left(RA\right)}\times Z_{IN}\times sC_{S\left(RA\right)}}\right)\times V_{S\left(RA\right)}-\left(\frac{Z_{P\left(LA\right)}\times Z_{IN}\times sC_{S\left(LA\right)}}{Z_{P\left(LA\right)}+Z_{IN}+Z_{P\left(LA\right)}\times Z_{IN}\times sC_{S\left(LA\right)}}\right)\times V_{S\left(LA\right)}$$
(13)$$V_{OUT}=\left(\frac{\frac{1}{Z_{P\left(RA\right)}}}{\frac{1}{Z_{IN}}+\frac{1}{Z_{P\left(RA\right)}}+sC_{S\left(RA\right)}}-\frac{\frac{1}{Z_{P\left(LA\right)}}}{\frac{1}{Z_{IN}}+\frac{1}{Z_{P\left(LA\right)}}+sC_{S\left(LA\right)}}\right)\times V_{EMI}\;+\left(\frac{sC_{S\left(RA\right)}}{\frac{1}{Z_{IN}}+\frac{1}{Z_{P\left(RA\right)}}+sC_{S\left(RA\right)}}\right)\times V_{S\left(RA\right)}-\left(\frac{sC_{S\left(LA\right)}}{\frac{1}{Z_{IN}}+\frac{1}{Z_{P\left(LA\right)}}+sC_{S\left(LA\right)}}\right)\times V_{S\left(RA\right)}$$

The (13) is summarized for $V_{OUT}$, which is a signal of the difference by the INA, and $V_{OUT}$, is the sum of the ECG signal and the EMI components.

As for the EMI component, if the magnitude of each impedance is equal, the same phase of signals is generated at each sensing electrode and can be removed by the INA. However, if the RA or LA electrode is partially detached away from the skin, each impedance is different, so the EMI components cannot be removed.

It is virtually impossible to keep each impedance with a constant value at every moment. However, if the size of the electrodes becomes larger, the capacitance ($sC_{S\left(RA\right)}$, $sC_{S\left(LA\right)}$) becomes sufficiently larger to be a dominant factor in the equation, so the influence on the difference in input impedance ($Z_{IN}$) and the difference in impedance between each electrode and the power line ($Z_{P\left(RA\right)}$, $Z_{P\left(LA\right)}$) can be neglected and the EMI components can fall to zero.

The ideal ECG signal is an intact output of the biopotential difference between $V_{S\left(RA\right)}$ and $V_{S\left(LA\right)}$ generated by each electrode. However, undesired signals can be inserted from the terms of V_EMI which consists of an impedance ($Z_{P}$) between power line and body interface, and a buffer input impedance ($Z_{IN}$). If $sC_{S}\;$is much larger than ($\frac{1}{Z_{IN}}+\frac{1}{Z_{P}}$) to avoid the attenuation caused by the $Z_{P}$ and $Z_{IN}$, then $sC_{S}$ becomes the dominant factor, so (13) can be rearranged as (14). The difference between $V_{S\left(RA\right)}$ and $V_{S\left(LA\right)}$ results in the intact output from the body.
(14)$$V_{OUT}=\left(\frac{0}{sC_{S\left(RA\right)}}-\frac{0}{sC_{S\left(LA\right)}}\right)\times V_{EMI}+\left(\frac{sC_{S\left(RA\right)}}{sC_{S\left(RA\right)}}\right)\times V_{S\left(RA\right)}-\left(\frac{sC_{S\left(LA\right)}}{sC_{S\left(LA\right)}}\right)\times V_{S\left(LA\right)}=V_{S\left(RA\right)}-V_{S\left(LA\right)}$$

If $sC_{S}$ is large enough to satisfy (14), the attenuation of the ECG signal and the EMI can be minimized.

### 2.4. Measurement Conditions of ECG Signals

Figure 5a shows a skintight garment and the position of the electrodes. The garment was utilized to make a secure contact between the electrodes and skin. The tight contact is necessary to avoid unwanted capacitance generated by the gap between the electrodes and skin. Additionally, each center of the different sized electrodes was fixed at the same position.

For different electrode sizes (diameters of 0.5, 1, 3, 5, 7, and 9 cm) and motion conditions (static and dynamic), ECG data was measured for 5 min from the same subject (male, 27 years). The ECG signals in this paper were measured using only one board in order to eliminate the influence of the element error and the state of the board. The data used in this paper are raw data without software processing, which can distort the ECG signal [35,36,37].

## 3. Results and Discussion

### 3.1. ECG Signal Amplitude Variation with Different Electrode Size

Figure 6 shows the ECG signals measured in a steady state according to different diameters of the electrodes. In graphs (a) to (f), the x−axis indicates time and the y−axis indicates a potential difference. The amplitude and the standard deviation (*n* = 30) of the QRS complex of each ECG signal are as shown in Figure 6a–f calculated by the difference between the average of the upper envelope and the average of the lower envelope. As shown in Figure 6g, with this calculation, amplitudes of the QRS complex were 1.08, 1.67, 1.52, 1.41, 1.49, and 1.47 mV with the diameter of 0.5, 1, 3, 5, 7, and 9 cm, respectively. Results show that the magnitude of the QRS complex is sharply increased when the diameter of the capacitive−coupled electrode is increased from 0.5 to 1 cm. In the case of the ECG signal measured with an electrode of 1 cm or more, even if the size of the electrode is increased, the amplitude is not increased any more, and only a slight amplitude fluctuation was observed within a certain range. It is presumed that the cause of the fluctuation is an error in the electrode center position.

When the size of the electrode is increased from 0.5 to 1 cm, it was determined that the cause of the sharp change in attenuation was that the capacitance of the electrode was not sufficient enough to store the charge on the skin. The above result is the same as the expectation in (12) that if $sC_{S}$ is not large enough compared to $\left(\frac{1}{Z_{IN}}+\frac{1}{Z_{P}}\right)$, the potential difference $\left(V_{S\left(RA\right)}-V_{S\left(LA\right)}\right)$ between the RA and LA electric fields may be attenuated.

### 3.2. EMI Influence with Different Diameter of Electrode

In general circumstances, the ECG measurement circuit and soldering pad are well electrically packaged. Therefore, EMI is not significant unless the capacitive−coupled electrodes are detached from the skin. However, if the skin–electrode contact is poor or exposed to high power electromagnetics, it can impair the ECG signal’s interpretation.

An experiment was conducted to verify the effect of EMI (power line frequency: 60 Hz) on the size of the electrodes. In the experiment, the sizes of the RA, LA, and DRL electrodes were increased with an additional floating electrode having a diameter of 3 cm connected to the RA electrode pad of the ECG measurement circuit. Then, we measured the amplitude change of the 60 Hz using fast Fourier transform (FFT) acquisition according to the size of the electrodes. Figure 7a–f show 30 s of time domain ECG signals in which 60 Hz components are mixed, and the signals were converted to frequency domain signals by FFT to compare the amplitude of the 60 Hz component according to the diameter of the electrode. These are represented in Figure 7g. As the size of the electrodes is increased, the 60 Hz component is decreased. When the size of the electrode is increased from 0.5 cm to 3 cm, a gradual change in the 60 Hz amplitude was observed. Particularly, when the diameter size of the electrode was increased from 0.5 cm to 1 cm, the 60 Hz amplitude was sharply decreased. This result is the same as the QRS signal interpretation in (13). When the impedance between the body and the power line is constant, the EMI is decreased as the source impedance of the skin–electrode interface is decreased. Based on the results, electrodes of at least 1 cm or more should be appropriated to avoid the EMI.

### 3.3. Baseline Wandering with Different Diameter of Electrode

ECG measurements were performed to check baseline wandering caused by the change in the gap between the electrode and skin in the ambulatory environment according to the size of the electrode. The movement was similar to the walking speed of 5 km/h, and the signals were measured while walking in place so that motion artifact would occur. Figure 8 shows ECG signals measured in a dynamic state according to different electrode sizes. In Figure 8a–f, the solid lines represent the ECG signals and the dotted lines represent the baseline fluctuation which was calculated using the moving average of the ECG signals. In the introduction, it was predicted that the larger size of the electrode makes the larger capacitance and the higher probability of the gap change between the electrode and skin. Figure 8g is a box plot of the amplitude comparison of the baseline wandering according to the sizes of the electrodes. As shown in Figure 8g, it was confirmed that as the size of the electrode is increased, the baseline wandering grows as well. Particularly, the magnitude of the baseline wandering is sharply increased at 3 cm electrode. Therefore, it is desirable to use an electrode size of 3 cm or less when measuring ECG in a dynamic environment to minimize the motion artifact. As analysis with the equivalent circuit, if $sC_{S}$ is sufficiently larger than $\frac{1}{Z_{IN}}+\frac{1}{Z_{P}}\;$and good contact between the electrodes in the body can be ensured using an electrode of 3 cm or less, it can be confirmed that the capacitive-coupled electrode is suitable for ambulatory ECG in real−life.

### 3.4. Effect of Sweat on ECG Measurement

Substances such as sweat from the body during ECG measurements can change the impedance between the electrodes and skin and produce slipping of the electrode. With previous experiment results, we chose a 3 cm diameter electrode because the electrode had good performance in amplitude, baseline wandering, and EMI. Figure 9 shows 6−min ECG signals measured during on−spot walking conditions with spraying saline on the skin to mimic the sweating environment. As shown in Figure 9a, some noise was generated at the baseline without baseline wandering and EMI due to the slipperiness between the skin and electrodes caused by the saline. As shown in Figure 9b, significant noise was not observed after 4 min. It is assumed that the moisture between the electrode and skin is vaporized, so the frictional force is increased over time. Additionally, even if some moisture still exists between the electrodes and skin, the amplitude and noise of the signals are not significantly different from the ECG signals measured in the dry state. This result shows that there is little change in impedance between the electrode and skin when using saline due to the high resistivity of silicon dioxide coating.

## 4. Conclusions

We analyzed the ECG signals with a capacitive−coupled electrode to optimize the area of the capacitive−coupled electrode for long term ambulatory ECG measurement system. The potential difference between the RA and LA, which was measured using the capacitive−coupled electrode, was obtained through a circuit having a filter with a cutoff frequency of 0.003 to 200 Hz, ADC, and Bluetooth module. The measured ECG signal was obtained according to various electrode sizes in a static state, an ambulatory state, and a noise state. A part of the electrode was detached from the skin when measuring the noise state.

The amplitude of the QRS complex of the ECG signals measured at a static state was calculated using the envelope of the signals. The amplitude of the QRS complex was remarkably increased from 1.08 mV to 1.67 mV when the diameter of the electrode was increased from 0.5 to 1 cm. There was no significant change in the amplitude when using 1 cm or more diameter electrodes. From this result, it was determined that the 0.5 cm electrode had insufficient capacitance to store the charge on the skin surface. The 60 Hz component of ECG signals measured with the 0.5 cm diameter electrode was more than five times larger than the 1 cm diameter electrode. As the area of the electrode is increased, the 60 Hz signal is relatively decreased, and the ECG signal was not impaired by EMI from the 3 cm or more diameter electrode. If the ECG circuit and capacitive−coupled electrode were well insulated from the power line or the earth ground, ECG signals could be obtained without 60 Hz by theoretical analysis of the circuit, even though the small electrode was utilized.

In order to check the baseline wandering of the ECG signal during dynamic movement, the ECG signal was measured using electrodes of various sizes while walking at a speed of 5 km/h. The baseline wandering of the signals was calculated by a moving average method. The experimental results confirm that the electrodes with diameters of 0.5 cm and 1 cm show similar levels of wandering and that the wandering becomes more intense from the electrode with a 5 cm diameter. The result shows that the electrode area is too large compared to the adhering electrode area, making it difficult to close contact with the electrode. That is, the gap changes more frequently than the small electrode.

In conclusion, the size of the electrode must be a minimum diameter of 1 cm or more to avoid attenuation of the signals and not be affected by EMI. To reduce baseline wandering in ambulatory ECG measurements, the diameter of the capacitive−coupled electrode must be less than 3 cm. Based on the circuit analysis and experiment results, if we utilize a silicon−dioxide−coated capacitive−coupled electrode within a diameter of 1 to 3 cm, signal attenuation, EMI effects, and motion artifacts can be minimized during real−life movement in ambulatory ECG measurements.

## Figures and Tables

**Figure 1 sensors-22-08388-f001:**
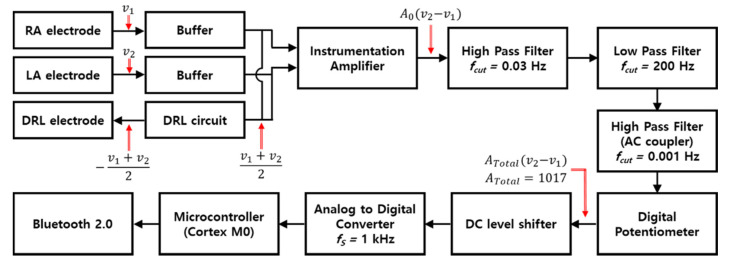
Block diagram of ECG measurement circuit.

**Figure 2 sensors-22-08388-f002:**
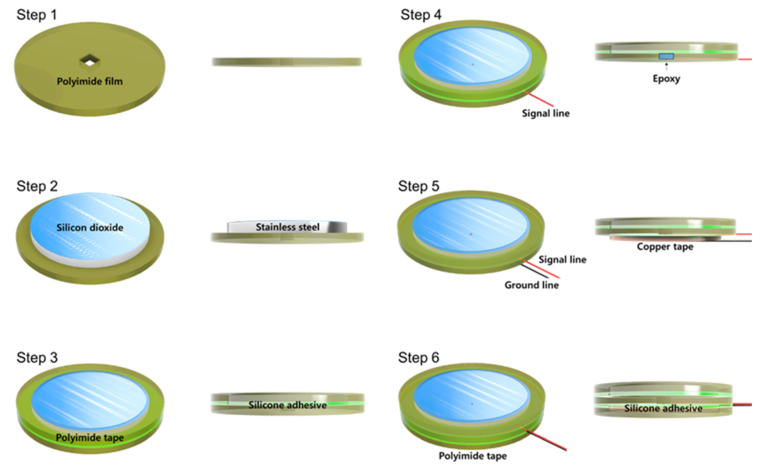
Fabrication process flow of silicon−dioxide−coated electrode.

**Figure 3 sensors-22-08388-f003:**
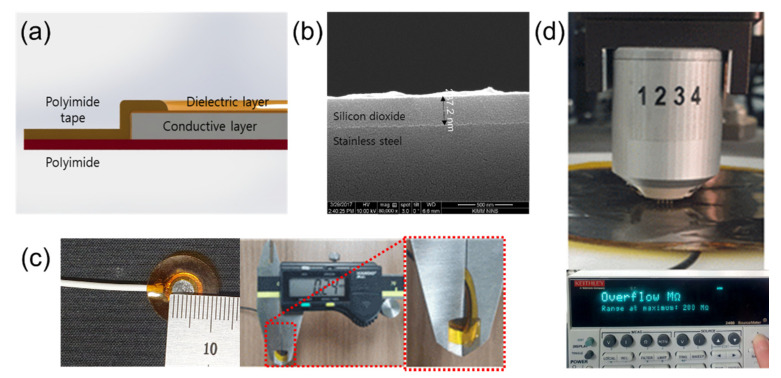
Silicon−dioxide−coated capacitive−coupled electrode. (**a**) Structure schematic of the capacitive−coupled electrode. (**b**) SEM (Fei company, Nova 200 NanoLab) picture of silicon dioxide. (**c**) Fabricated electrode (diameter: 0.5cm) and total thickness (240 μm). (**d**) Surface resistance measurement of capacitive−coupled electrode using 4−point probe.

**Figure 4 sensors-22-08388-f004:**
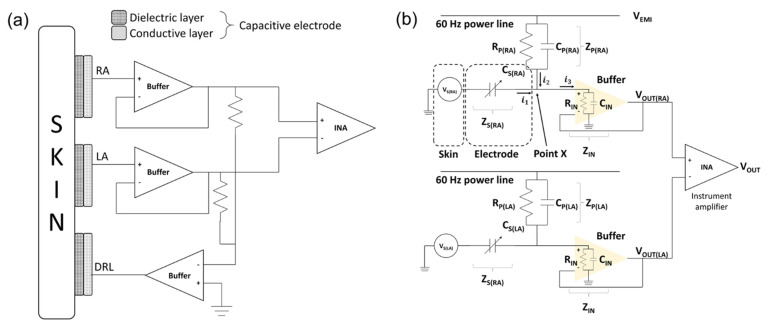
Schematic of skin–electrode interface. (**a**) Mechanical connection between electrode and circuit and (**b**) electrical circuit model of skin–electrode interface.

**Figure 5 sensors-22-08388-f005:**
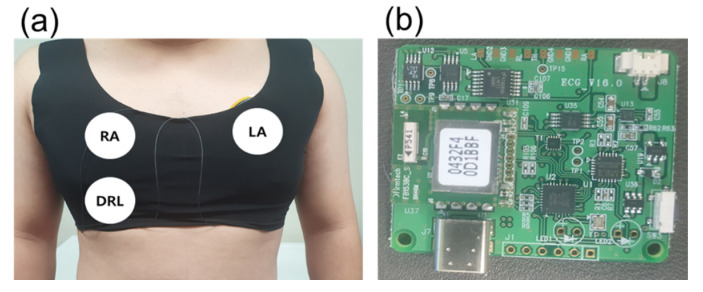
Measurement environment. (**a**) Location of capacitive electrode and (**b**) ECG measurement board.

**Figure 6 sensors-22-08388-f006:**
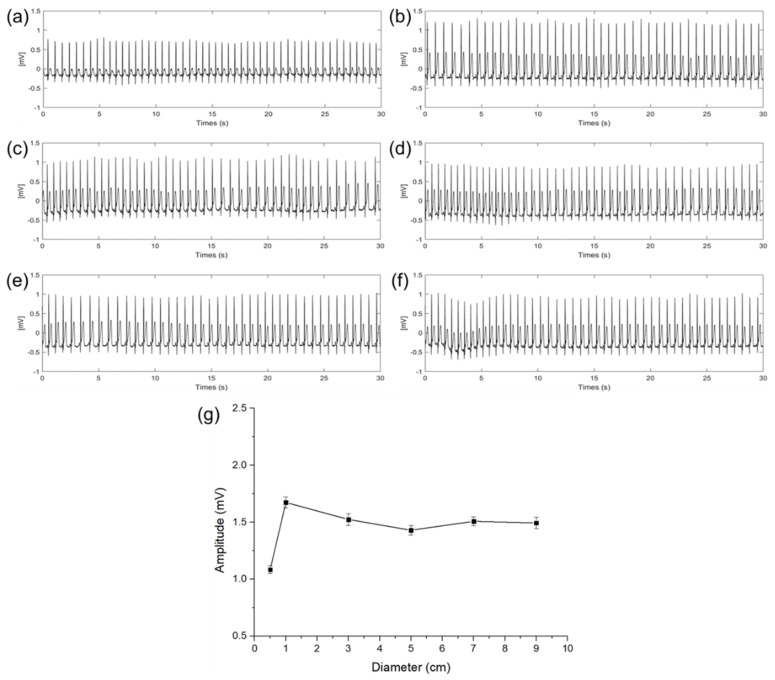
ECG signals of electrode diameter with (**a**) 0.5 cm, (**b**) 1 cm, (**c**) 3 cm, (**d**) 5 cm, (**e**) 7 cm, (**f**) 9 cm, and (**g**) QRS complex amplitude and standard deviation comparison according to the diameter of electrodes.

**Figure 7 sensors-22-08388-f007:**
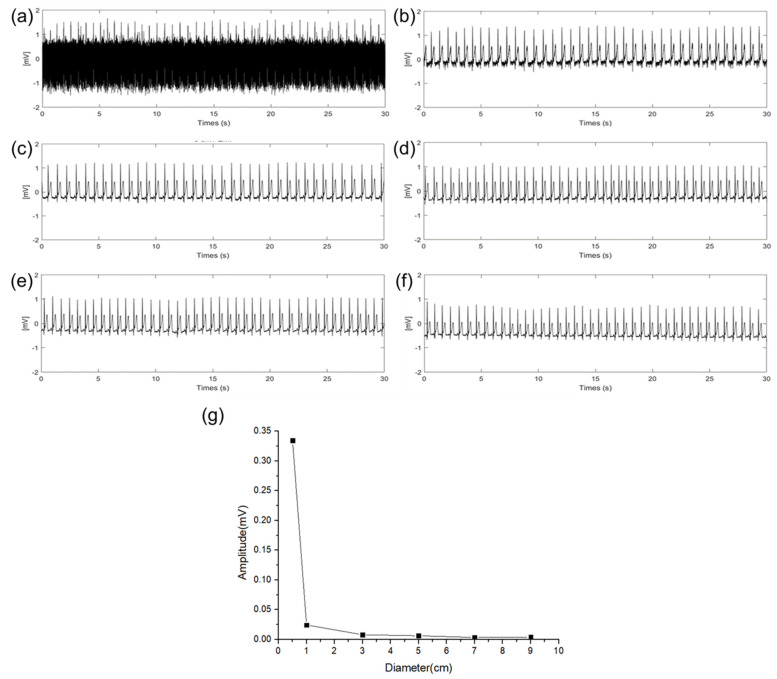
EMI influence according different electrode diameter of (**a**) 0.5 cm, (**b**) 1 cm, (**c**) 3 cm, (**d**) 5 cm, (**e**) 7 cm, (**f**) 9 cm, and (**g**) EMI influence comparison according to the diameter of electrodes.

**Figure 8 sensors-22-08388-f008:**
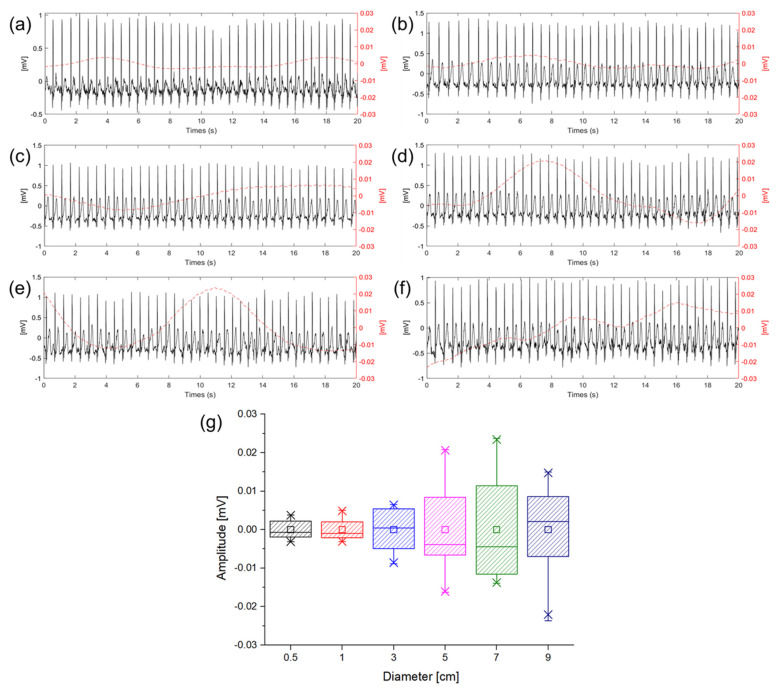
Baseline wandering according different electrode diameter of (**a**) 0.5 cm (black), (**b**) 1 cm (red), (**c**) 3 cm (blue), (**d**) 5 cm (violet), (**e**) 7 cm (green), (**f**) 9 cm (indigo), and (**g**) box plot of baseline wandering amplitude according to the diameter of electrode.

**Figure 9 sensors-22-08388-f009:**
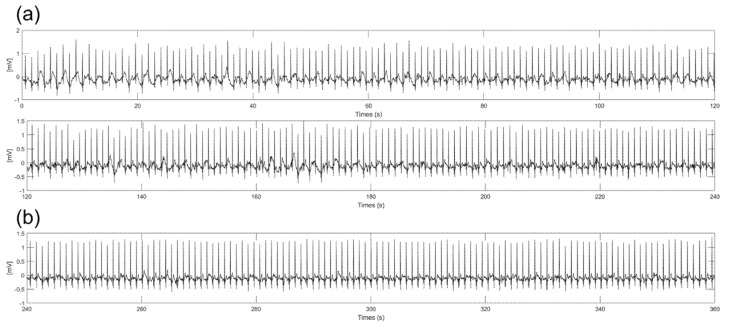
Six−minute ECG signals with spraying the saline on the skin. (**a**) From start to 4 min (**b**) From 4 to 6 min.

## Data Availability

Not applicable.

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
