# Peer review of "Development and Optimization of Silicon−Dioxide−Coated Capacitive Electrode for Ambulatory ECG Measurement System"

_sensors, 2022, doi:10.3390/s22218388_

Round 1
Reviewer 1 Report
In this paper, the authors developed and optimized a silicon dioxide-coated capacitive electrode for ambulatory electrocardiogram (ECG) measurement system. The study consists of experimental work and explores an innovative method for measuring the ECG signals with good accuracy and repeatability. The manuscript is professionally written with minimal grammatical and typological errors and having clear and legible figures. There are very minimal errors highlighted in yellow in the uploaded document with comments attached them. These are:
(1) In the introduction: (a) in line 69 it is suggested to add the unit to the number 166. Is it in um and (b) in line 84 it is suggested to replace “to on” with “on to”.
(2) In ECG Measurement System and Measurement: (a) line 153 it is suggested to write all variables in italic fonts and (b) in line 294 it is suggested to delete the word “is”.

Author Response
Response to Reviewer 1 Comments
Point 1:
In the introduction: (a) in line 69 it is suggested to add the unit to the number 166. Is it in um and (b) in line 84 it is suggested to replace “to on” with “on to”.
Response 1:
The unit of the number 166 is “layer”. And to clearly describe the meaning of sentences, we have revised the sentence from
“166 thin alternative Al2O3 / TiO2 layers were deposited by atomic layer deposition (ALD) on the conductive layer which is attached to the polyimide as a substrate of the electrode.”
to (in lines 68 - 70)
“A stack of 166 alternate layers of Al2O3 and TiO2 were deposited by atomic layer deposition (ALD) on the conductive layer which is attached to the polyimide as a substrate of the electrode.”
And replace “to on” with “on to” in line 83.
Point 2:
In ECG Measurement System and Measurement: (a) line 153 it is suggested to write all variables in italic fonts and (b) in line 294 it is suggested to delete the word “is”.
Response 2:
We change the fonts of all variables of the sentence in line 156
“In (1), ε0 is the dielectric constant of free space, εr is the dielectric constant of silicon dioxide (εr = 4), d is the thickness of the silicon dioxide layer (d = 287 nm), and A is the area of each electrode.”
And the sentence in line 298 is changed as shown below
“the 60 Hz amplitude was sharply decreased.”

Reviewer 2 Report
1. This paper deals with the development and optimization of Silica Dioxide-coated capacitive electrodes for ECG measurements. The features of the silica dioxide-coated capacitive electrode, such as the uniformity, the size of silica in the electrode coating and so on , should be highlighted in the paper, but these important aspects are missing.
2. The novelty of this article is the use of a silica dioxide coating as a dielectric layer between the electrode and the skin, which is interesting. However, the authors did not compare their electrodes with existing ones.
3. Factors that affect electrode performance include material, thickness, and area. However, the authors only analyze influence of the electrode area, and neither explain the reason for choosing silica nor analyze the influence of thickness.
4. This sentence “Whole materials were prepared through laser machining” is a bit weird.
5. Some figures are not clear,such as Fig.3(c)
Author Response
Response to Reviewer 2 Comments
Point 1:
This paper deals with the development and optimization of Silica Dioxide-coated capacitive electrodes for ECG measurements. The features of the silica dioxide-coated capacitive electrode, such as the uniformity, the size of silica in the electrode coating and so on, should be highlighted in the paper, but these important aspects are missing.
Response 1:
The proposed electrode in this paper measures an ECG using the capacitive coupling effect by silicon dioxide-coated electrodes which have a large resistivity. The resistance of the coated silicon dioxide layer was measured as a sufficiently large value of over 200 Mohm, so we interpreted the silicon dioxide layer as a single capacitor, not a parallel connection of resistor and capacitor. We think the characteristics such as the uniformity, and thickness of the coating layer are not factors that significantly affect the experimental results because the thin thickness of the electrode has sufficient flexibility to follow the curvature of the human body and good skin contact for ECG measurement.
Point 2:
The novelty of this article is the use of a silica dioxide coating as a dielectric layer between the electrode and the skin, which is interesting. However, the authors did not compare their electrodes with existing ones.
Response 2:
Other capacitive electrode types were introduced in lines 63 – 74. And the merits of our silicon dioxide-coated electrode were explained in lines 89 – 93.
Point 3:
Factors that affect electrode performance include material, thickness, and area. However, the authors only analyze the influence of the electrode area, and neither explains the reason for choosing silica nor analyze the influence of thickness.
Response 3:
The reasons for choosing silicon dioxide as a dielectric layer were explained in lines 89 – 93. To confirm the influence of the impedance difference among the various electrodes, we control the area of the electrode. Considering the mechanical characteristics of the skin-electrode interface, the thickness of the silicon dioxide layer is not an important factor in the analysis of the signal which is conducted in this paper. And the analysis influence of the thickness will be conducted in the next paper with other signal analysis methods and experiment environments.
Point 4:
This sentence “Whole materials were prepared through laser machining” is a bit weird
Response 4:
We have revised the sentence from
“Whole materials were prepared through laser machining.”
to(in line 131)
“Whole layers of the electrode were cut through laser machining.”
Point 5:
Some figures are not clear,such as Fig.3(c)
Response 5:
We have replaced the Figure. 3(c) and it is included in PDF file.

Reviewer 3 Report
This paper prepares a silicon dioxide-coated capacitive electrode for ambulatory ECG measurement.
My remarks for consideration:
- Wearable electrode is suitable for long-term ECG monitoring. Moreover, it can reduce baseline wander noise and motion artifact because of good flexible property, which makes it get good contact between the electrode and the skin. The related work are missing in this manuscript. Please cite the following papers, such as
DOI:10.1007/s12274-021-3536-3
DOI:10.1021/acsami.1c23567
- The electrode has a very high surface resistance of over 200 MΩ. However, we focus on the contact impedance between the electrode and skin for ambulatory ECG measurement. The research should discuss the problem.
- How to make sure each center of the different electrodes fixed at the same position? What’s the same position? Body or garment?
Author Response
Response to Reviewer 3 Comments
Point 1:
Wearable electrode is suitable for long-term ECG monitoring. Moreover, it can reduce baseline wander noise and motion artifact because of good flexible property, which makes it get good contact between the electrode and the skin. The related work are missing in this manuscript. Please cite the following papers, such as
DOI:10.1007/s12274-021-3536-3
DOI:10.1021/acsami.1c23567
Response 1:
We have revised the sentence from
“The capacitive coupled electrodes were fabricated with thin and flexible conditions so that they would adhere well against the curvatures of the skin.”
to(in lines 159 – 163)
“The capacitive coupled electrodes were fabricated with thin and flexible conditions to adhere well against the skin curvatures, so it guarantees good mechanical contact between the skin and the electrode and this characteristic reduces baseline wander and motion artifacts [29,30]”
And, We have cited the paper you recommended.
Point 2:
The electrode has a very high surface resistance of over 200 MΩ. However, we focus on the contact impedance between the electrode and skin for ambulatory ECG measurement. The research should discuss the problem.
Response 2:
In line 168, we have described that the resistance was measured by the 4-point probe and the value was over 200Mohm. In line 188, we describe the skin-electrode interface as only working as a single capacitor, not a parallel connection of resistor and capacitor. The electrode is directed contact with flexible skin. Therefore, the contact (real part) between the electrode and skin can be neglectable.
Point 3:
How to make sure each center of the different electrodes fixed at the same position? What’s the same position? Body or garment?
Response 3:
"Same position" means the center location of the contact site of the body and electrode, not the garment. Various sizes of electrodes were used in consecutive order by size. After measuring the ECG signal with the bigger size of the electrode, we indicate the outline of the chest and aligned the smaller electrode on the indicated outline.

Reviewer 4 Report
The article “Development and Optimization of Silicon Dioxide-Coated Capacitive Electrode for Ambulatory ECG Measurement System” presents a silicon dioxide-coated capacitive electrode system for an ambulatory electrocardiogram (ECG). The 240 um thickness electrode was composed of a stainless-steel sheet layer for sensing, a polyimide electrical insulation layer, and a copper sheet connected with the ground to block any electrical noises generated from the back side of the structure. Six different diameter electrodes were prepared to optimize ECG signals in ambulatory environment, such as the amplitude of the QRS complex, magnitude of electromagnetic interference (EMI), and baseline wandering of the ECG signals.
This paper demonstrates that the size of the electrode must be a minimum diameter of 1 cm or more to avoid attenuation of signals and not be affected by EMI. To reduce baseline wandering in ambulatory ECG measurements, the diameter of the capacitive coupled electrode must be less than 3 cm.
This paper is logically organized and clearly structured. The purpose and significance of the study are clearly stated and the research method is appropriate because it is based on an experimental measurement on six different electrodes on real case. This is an important aspect of the paper because the experimental data examples provide by the authors is quite interesting. The experimental work is very well explained and brings a lot to the understanding of the research.
However, several points should be enhanced.
First one is the electrical characterization of the electrodes. In the abstract, the authors claim that the resistance of silicon dioxide layer is around 200 Mohms. No measurement are performed, so it should be very important to apply each electrode on a network analyzer to obtain the real and imaginary part of the impedance.
Second point, Figure 1 is not a block diagram of the ECG measurement circuit, it is more a wiring diagram. So a real block diagram should be added with the main function, buffering, filtering, DC level shifter, AD conversion, wireless connection, etc… Something is missing between ADC and Bluetooth, where is the microcontroller?
Third item concerns the soldering aspects. How it is done? Conducting compound? What is the thickness? It should have an effect on the electrical characteristics of the sensors.
In Part 3 “Results and Discussions” , it should be interesting to present a figure of the definition of the QRS signal because QRS amplitude is higher than the peak value of the ECG signal.
The corrections concern the presentation of the paper. The list of improvements is as follow:
- Line 12: How is obtain the 200 Mohms value, experimental or theoretical?
- Figure 1: RA, LA and DRL electrode should be defined here.
- Line 154: Measured impedances of the electrodes should appear here.
- Line 172, DRL electrode should be explained inside the text
- Line 182: is it the real part of the impedance?
- Line 264: How can you prove it?
- Line 286: No frequency distribution on the figure 6
- Fig 7: Is Figure g the Power Spectral Density of the spectrum of the time signals? It should appears. Magnitude is not the good term.
-
-
Despite some corrections have to be done, the paper presents an original work which can be published in Sensors Journal after minor revisions.

Author Response
Response to Reviewer 4 Comments
All figures are attached in PDF file.
Point 1:
First one is the electrical characterization of the electrodes. In the abstract, the authors claim that the resistance of silicon dioxide layer is around 200 Mohms. No measurement are performed, so it should be very important to apply each electrode on a network analyzer to obtain the real and imaginary part of the impedance.
Response 1:
The “200 Mohms” in abstract is the measured value by 4-point probe. The measurement is explained in line 163-168.
Point 2:
Second point, Figure 1 is not a block diagram of the ECG measurement circuit, it is more a wiring diagram. So a real block diagram should be added with the main function, buffering, filtering, DC level shifter, AD conversion, wireless connection, etc… Something is missing between ADC and Bluetooth, where is the microcontroller?
Response 2:
We replace the block diagram in Figure 1 as shown below. And also add the MCU, digital potentiometer, and the output signals of the remarkable block.
Figure 1. Block diagram of ECG measurement circuit
Point 3:
Third item concerns the soldering aspects. How it is done? Conducting compound? What is the thickness? It should have an effect on the electrical characteristics of the sensors.
Response 3:
We soldered the wires to stainless steel by lead-free solder, ultrasonic soldering iron, and stainless steel flux. And the thickness of the soldering spot mounds was almost under 1 mm. After soldering, we inspect the electrical connection between the substrate and the opposite side of the wire with a digital multimeter whether the value is below 0.1 ohms.
Point 4:
In Part 3 “Results and Discussions”, it should be interesting to present a figure of the definition of the QRS signal because QRS amplitude is higher than the peak value of the ECG signal.
Response 4:
The value of the QRS amplitude which is used in Part 3 "Result and Discussions" was induced in the figure below. We converted 1017 times amplified analog ECG signals in the 0-3.3 V range to digital signals using an ADC with 10-bit resolution. Decimallized raw data received through Bluetooth is (a) in the figure below. Figure (b) shows the converted data from raw data into millivolts (mV) by the equation: (raw data)/(1023*3.3*1000). In this paper, we analyzed the cardiac conduction system as AC coupled system and subtract the mean value of the (b) from the data shown in Figure (b). The result of the calculation is shown in Figure (c) and the offset of the signal would be 0. In each figure, the mark 'X' denotes the R wave and the mark 'O' denotes the S wave of the ECG signal. In Fig. (c), the red line located on the upper side of the signal is the average value of the R waves, and the blue line located on the lower side of the signal is the average value of the S waves. The value indicated as QRS amplitude in this paper is calculated by the equation as follows (the average voltage of R waves- average voltage of S waves). The S waves in this paper are always a negative voltage, so the QRS amplitude may be higher than the peak value (R wave) of the ECG signals.
Point 5:
Line 12: How is obtain the 200 Mohms value, experimental or theoretical?
Response 5:
200 Mohms was experimentally measured with a 4-point probe(Keithley, 2400 source meter). And it was not exactly 200 Mohms. The limitation of the 4-point was 200 Mohm, so we assume that the surface resistance of the electrode was over 200 Mohms. The measuerment was explained from line 163-168.
Point 6:
Figure 1: RA, LA and DRL electrode should be defined here.
Response 6:
We have revised the sentence from
“As shown in Figure 1, the block diagram of an ECG measurement circuit is composed of buffers, a driven right leg (DRL) circuit, an instrumentation amplifier (INA), a band-pass filter, an AC coupling circuit, a DC level shifter, an analog to digital converter (ADC), and a Bluetooth module.”
to (in line 103-110)
“Figure 1 is the block diagram of an ECG measurement circuit. The circuit and human body are connected with 3 electrodes. The right arm (RA) and left arm (LA) electrode is attached at the right side of chest and left side of the chest, respectively, and the driven right leg (DRL) electrode is attached to the right leg to transmit the common-mode signal which is generated by DRL circuit to the body. And the circuit is composed of buffers, a DRL circuit, an instrumentation amplifier (INA), a band-pass filter, an AC coupling circuit, a digital potentiometer, a DC level shifter, an analog to digital converter (ADC), a microcontroller (cortex M0), and a Bluetooth module.”
Point 7:
Line 154: Measured impedances of the electrodes should appear here.
Response 7:
In this paper, we did not measure the impedance of the electrode. As you may know, the area of the electrode and the impedance are inversely proportional, and we analyzed the ECG signal by changing the area of the electrode to observe the tendency, not an exact value. In line 188, we describe the skin-electrode interface as only working as a single capacitor, not a parallel connection of resistor and capacitor. That is, the real part(the resistor) can be removed as shown in Fig4. Therefore only capacitor is afftect to the result. And the capacitance is usually varied by area if you have same material and almost same gap between electrode and skin. If we have high resistive electrode, the impedance maearurement is not a crutial point.
Point 8:
Line 172, DRL electrode should be explained inside the text
Response 8:
The meaning of LA, RA, and DRL was explained in line 104. So, we have revised the sentence from
“The ECG signals were measured by a bipolar limb lead. The electrodes are consisted of a right arm (RA) electrode, a left arm electrode (LA), and a DRL electrode.”
to in lines (175 - 176)
“The ECG signals were measured by a bipolar limb lead. The electrodes have consisted of RA, LA, and DRL electrodes.”
Point 9:
Line 182: is it the real part of the impedance?
Response 9:
Yes. The value was measured by 4-point probe.
Point 10:
Line 264: How can you prove it?
Response 10:
Additional experiments were conducted to prove that the reason why the QRS complex of the signal measured using an electrode with a diameter of 0.5 cm is attenuated. Electrodes with diameters of 0.5, 0.75, and 1 cm were fabricated, respectively, and the same experiment was performed, and the results of the experiment are shown below. In several experiments, as the diameter of the electrode increased from 0.5 cm to 1 cm, it was observed that the amplitude of the QRS complex also increased proportionally to the diameter of the electrode as shown in the following figure. Thus we considered that the 0.5cm is not enough diameter to sotre the charge generated from cadiac conduction system.
In our system, the charge storing is saturated around 1cm electrode.
Point 11:
Line 286: No frequency distribution on the figure 6
Response 11:
We have revised the sentences as shown below because of their ambiguous.
From
“The left side of Figure 7 (a) to Figure 7 (f) are ECG signals in which 60 Hz components are mixed, and the right side of the figures are graphs of the frequency distribution of the mixed signal.
Time domain signals of 30 seconds in Figure 7 (a) to Figure 7 (f) were converted to frequency domain signals by FFT to compare the magnitude of the 60 Hz component according to the diameter of the electrode.”
to in lines (289 – 292)
“Figure 7 (a) to Figure 7 (f) show 30 seconds of time domain ECG signals in which 60 Hz components are mixed. And the signals were converted to frequency domain signals by FFT to compare the amplitude of the 60 Hz component according to the diameter of the electrode.”
Point 12:
Fig 7: Is Figure g the Power Spectral Density of the spectrum of the time signals? It should appears. Magnitude is not the good term.
Response 12:
Fig 7. (g) is the amplitude of 60 Hz component. We have replace the word “magnitude” to “amplitude” in 3.2. EMI Influence with Different Diameter of Electrode., and line 18.
And we have revised the y axis of Fig 7.(g). as shown below. The draft version of Fig 7.(g). had been error in unit and y axis converting.

Round 2
Reviewer 2 Report
The authors responded to my comments, but some responses are questionable or unsatisfactory. For example, the author wrote “the analysis influence of the thickness will be conducted in the next paper ...”. Why in the next paper, not this one?
Author Response
Response to Reviewer 2 Comments
Point 1:
The authors responded to my comments, but some responses are questionable or unsatisfactory. For example, the author wrote “the analysis influence of the thickness will be conducted in the next paper ...”. Why in the next paper, not this one?
Response 1:
The authors would like to thank you for the additional comments. In this paper, we focused on silicon dioxide for the dielectric material between the skin and the electrode. The coated silicon dioxide has high resistance described in lines 165-168. Unlike conventional capacitive electrodes which are analyzed as parallels R and C, the proposed ECG electrode is operated as a single capacitor due to the high resistive property. With the pure capacitive coupled electrode, we have studied the performance of the ECG system with changing capacitance. This is the main purpose of this paper. As you know, capacitance can be controlled by permittivity, area, and gap. In this paper, with fixed material, we have two choices (gap and area) to change the capacitance. In the parameters, the capacitance variation described in line 158 by changing electrode area has a 330 times difference. To achieve the same difference of capacitance with the designed 287 nm thickness of dielectric material, the gap or thickness of dielectric material should be under 0.86 nm or over 94 um which is not easy to achieve in these day’s technology. However, we also think it is necessary to further study the thickness that can affect the durability in conditions under sweating or friction with skin according to the thickness of the coated layer.
